# Advances in Novel Animal Vitamin C Biosynthesis Pathways and the Role of Prokaryote-Based Inferences to Understand Their Origin

**DOI:** 10.3390/genes13101917

**Published:** 2022-10-21

**Authors:** Pedro Duque, Cristina P. Vieira, Jorge Vieira

**Affiliations:** 1School of Medicine and Biomedical Sciences (ICBAS), Porto University, Rua de Jorge Viterbo Ferreira, 228, 4050-313 Porto, Portugal; 2Faculdade de Ciências da Universidade do Porto (FCUP), Rua do Campo Alegre, s/n, 4169-007 Porto, Portugal; 3Instituto de Investigação e Inovação em Saúde (I3S), Universidade do Porto, Rua Alfredo Allen, 208, 4200-135 Porto, Portugal; 4Instituto de Biologia Molecular e Celular (IBMC), Rua Alfredo Allen, 208, 4200-135 Porto, Portugal

**Keywords:** ascorbic acid, evolution, synthesis, aldonolactone oxidoreductases, insects, nematodes, prokaryotes

## Abstract

Vitamin C (VC) is an essential nutrient required for the optimal function and development of many organisms. VC has been studied for many decades, and still today, the characterization of its functions is a dynamic scientific field, mainly because of its commercial and therapeutic applications. In this review, we discuss, in a comparative way, the increasing evidence for alternative VC synthesis pathways in insects and nematodes, and the potential of myo-inositol as a possible substrate for this metabolic process in metazoans. Methodological approaches that may be useful for the future characterization of the VC synthesis pathways of *Caenorhabditis elegans* and *Drosophila melanogaster* are here discussed. We also summarize the current distribution of the eukaryote aldonolactone oxidoreductases gene lineages, while highlighting the added value of studies on prokaryote species that are likely able to synthesize VC for both the characterization of novel VC synthesis pathways and inferences on the complex evolutionary history of such pathways. Such work may help improve the industrial production of VC.

## 1. Introduction

Vitamin C (VC) is a water-soluble compound that, at physiological pH, is mainly found in a reduced ionizable form (L-ascorbate) [1]. The potential applications of VC have been the subject of research since it was first synthesized in 1933 [2] to the present day, in many areas relevant to mankind, such as human nutrition [3,4], animal feed supplementation [5,6], agricultural and industrial waste management [7,8], and sustainable nanomaterial synthesis [9]. Recently, this nutrient gained further relevance as a potential therapeutic agent for COVID-19 prevention and treatment [10,11].

VC plays a role in a remarkable number of metabolic processes across the life kingdoms, but is more commonly known for its crucial functions in oxidative stress response, acting as a primary scavenger of reactive oxygen and nitrogen species which originated from exogenous (e.g., alcohol ingestion) and endogenous (e.g., mitochondrial metabolism) sources [12,13,14,15,16]. In this process, reactive species such as superoxide and hydroxyl radicals are reduced by VC to form one electron oxidized monodehydroascorbate (MDHA), which can be converted to fully oxidized nonionic dehydroascorbate (DHA) by interaction with other MDHA molecules in solution [1,13,17]. The subsequent oxidation steps of VC do not deplete the available VC pool in cells, as this nutrient can be regenerated from either MDHA or DHA using a combination of enzymatic and non-enzymatic sparing mechanisms in plants, animals, and fungi [16,17,18,19], or directly as a by-product of MDHA dismutation [16,17]. However, VC can be irreversibly hydrolyzed to 2,3-diketo-L-Gulonic acid if not reduced in a timely manner by a Glutathione-dependent system [16,20,21]. VC oxidation with consequent formation of MDHA was also found to be responsible for the partial recycling of the lipid-soluble Vitamin E [22,23], an important membrane antioxidant known to act directly against fatty acid peroxyl radicals, thus preventing cell and organelle membrane damage [24,25,26].

VC is a prerequisite for normal neuromodulation and overall protection against neurodegeneration in vertebrate species [27,28,29,30,31], as well as hepatic and renal protection [32,33,34,35]. Furthermore, VC seems to be a key element in cancer prevention and treatment, as the suppression of free radical accumulation can ultimately retard or avoid the development of tumors [36,37,38,39]. Paradoxically, VC can act as a pro-oxidant when present in high concentration in the cellular environment [40], a chemical feature that can also be beneficial in potential cancer-treatment therapies [41,42,43]. In this context, the VC-induced oxidative stress appears to induce cancer cell apoptosis by itself, or synergistically in combination with other therapeutic agents [44,45,46]. This effect might not be, however, tumor-specific [47,48]. Nevertheless, since a very reduced number of patients treated with high intravenous VC concentrations have side effects [49], such treatment appears to be safe from a human health perspective. VC is also important in collagen biosynthesis, acting as a cofactor for the collagen stabilization enzymes prolyl and lysyl hydroxylases [50,51]. Thus, adequate levels of this nutrient are required to avoid the degradation of connective tissues in animal species [52,53]. This molecule has also been implicated in the modulation of epigenetic modifications such as DNA and histone demethylation, acting as a regulatory cofactor of ten-eleven translocation dioxygenases and Jumonji C domain-containing histone demethylases [54,55], and in the facilitation of non-transferrin-bound iron uptake [56,57].

Metazoa and fungi share a close evolutionary relationship and represent the two major groups of the eukaryotic Opisthokonta [58,59]. Nevertheless, in fungi, knowledge on VC functions is relatively scarce. Fungi species mainly rely on five-carbon VC analogs for optimal metabolic function [60,61]. The most common, D-Erythroascorbate, seemingly displays antioxidant behavior [60]. Nevertheless, the supplementation of VC can also improve fungal enzymatic antioxidant system activity and reduce the overall oxidative stress induced by mechanical lesions, when in moderation [62]. Furthermore, exogenous VC was also shown to confer considerable protection against free radical species originating from paraquat exposure in *Saccharomyces cerevisiae* [63]. In *Sclerotinia minor*, a fungi species where endogenous VC synthesis may occur, this nutrient appears to play a role in sclerotium biogenesis [64], and thus is likely involved in the adaptation of this species to environmental stresses.

In plants, VC plays an active role on overall stress perception and response [65,66,67,68]. Indeed, VC has been shown to ameliorate the negative effects of high salinity and heavy metal exposure in agricultural crops [67,69,70,71,72]. The protective aspects of VC are mostly relevant to the apoplast, a plant matrix considered to be the primary target of external abiotic or biotic factors in the roots [73,74], the chloroplast [75,76], and the organelle, where important physiological processes such as photosynthesis and lipid metabolism occur [77]. In simplistic terms, in response to stress factors that induce oxidative stress, such as exposure to ozone, VC can be released from the cytoplasm of plant cells to the apoplast, through the plasma membrane, where it becomes enzymatically or non-enzymatically oxidized, while scavenging radical species [73]. In the apoplast, where the conditions are considerably acidic (pH between 5 and 6) and dehydroascorbate reductase is absent, the originated MDHA is very unstable and the capacity to regenerate VC from DHA is impaired, and thus the main metabolite that is present is DHA [73,75]. Although the research is limited, this high concentration of DHA can potentially interfere in relevant physiological mechanisms, such as the regulation of protein activity in the plasma membrane and cell wall [73]. To maintain a regulated VC/DHA ratio in the apoplast, DHA diffuses to the cytoplasm and is reduced back to VC [73,75]. In the chloroplast, VC intervenes as an electron donor for ascorbate peroxidases, which regulate oxidative stress by reducing hydrogen peroxide that results from the activity of the photosystems I and II through the “water-water cycle” [78,79,80,81,82,83]. VC is also a cofactor in the conversion of violaxanthin to zeaxanthin, catalyzed by violaxanthin de-epoxidase during the xanthophyll cycle, contributing to the protection of the photosynthetic apparatus from photodamage in excess light conditions [84,85,86].

Regarding prokaryotes, overall, VC is reported as not being required for normal metabolic functions [17]. Nevertheless, there is evidence for endogenous VC synthesis in these organisms [87,88,89,90,91,92,93], and some authors reported that VC can be decomposed and utilised for bacteria growth, particularly in the absence of other carbohydrate sources such as glucose [92]. Furthermore, the presence of VC also enables oxygen uptake in some *Bacterium* species [92], suggesting a protective role against increased oxidative stress. Therefore, the importance of VC for these basal organisms, might be more important than previously thought.

The current knowledge on VC molecular roles is likely to increase in the future, particularly as a way to further explore potential applications for human health and agriculture. Detailed knowledge on VC synthesis, in a wide variety of organisms, is also needed to explore such applications. Therefore, in the following sections, endogenous VC synthesis pathways are reviewed, with emphasis on animal species once thought to be unable to synthesize or even require this nutrient. In addition, the possibility of endogenous VC synthesis in prokaryotes will be addressed.

## 2. Vitamin C Biosynthetic Pathways: The Current Status

Although the origins of the divergent VC synthesis mechanisms in eukaryotes have not been completely elucidated, three main metabolic pathways have been well-described in photosynthetic protists, algae, plants and animals [94,95,96,97,98,99,100,101]. In addition, fungi species are known to synthesize VC analogs, such as D-erythroascorbate [102]. Due to the comprehensive reviews on the subject [17,68,80,94,103], and lack of significant recent advances, these will be only briefly discussed to highlight the main differences between pathways.

One of the distinctive features of the main VC synthesis pathways is the sugar precursor. Therefore, the main VC synthesis pathways are shown, not according to taxonomic relationships, but in order to highlight this point, as well as the sharing of the last enzyme of the pathway between photosynthetic protists and plants and green algae, and the sharing of the first enzyme of the animal and photosynthetic protists pathways (Figure 1). While plants rely mainly on D-Mannose as the main substrate for VC synthesis, animals and photosynthetic protists utilize D-Glucose [17]. Starting from different sugars, and by the use of divergent enzymatic mechanisms, plants and animals convert these sugars to L-Galactono-1,4-lactone and L-Gulono-1,4-lactone in the penultimate step of the pathway, respectively [17]. In photosynthetic protists, however, VC synthesis relies on a mix of animal and plant enzyme orthologs to confer the hybrid capacity for the conversion of D-Glucose to L-Galactono-1,4-lactone [17]. L-Galactono-1,4-lactone is ultimately metabolized to VC by the action of a L-Galactono-1,4-lactone dehydrogenase (GALDH) in plants and photosynthetic protists, while in animals, L-Gulono-1,4-lactone originates VC through an oxidation step catalyzed by L-Gulonolactone oxidase (GULO) [17]. It should be noted that some of the intermediate sugar substrate conversion steps in the animal and photosynthetic protists pathways are still hypothetical, since the putative enzymes involved in these steps have not yet been isolated and functionally characterized. For instance, the UDP-Glucuronidase that catalyzes the transition of UDP-D-Glucuronate to D-Glucuronate is thought to exist in vertebrate species but has not yet been identified [104]. Fungi species rely on the five carbon D-Arabinose sugar to synthesize D-erythroascorbate after two conversion steps catalyzed by D-Arabinose dehydrogenase and D-Arabinono-1,4-lactone oxidase (ALO), respectively [17].

In plants, alternative mechanisms in which myo-Inositol, L-Gulose and D-Galacturonate are used as precursors of VC synthesis, have been proposed [105,106,107]. They are based on species-specific observations, and they have not been yet verified in a wider taxonomic context, and thus are not here discussed.

## 3. Alternative Animal VC Synthesis Pathways

The pseudogenization or even complete loss of the *GULO* gene is often reported as the primary cause of VC synthesis inability, in species such as humans, teleost fish, insects and nematodes [109,110,111,112,113]. Nevertheless, recent findings challenged the idea that the absence of a functional *GULO* gene can be used as a marker of VC synthesis inability in all animals [114,115,116]. In addition, although scarcely mentioned, the notion that D-Glucose is exclusively used for animal VC synthesis has also been challenged [117].

### 3.1. VC Synthesis in Caenorhabditis Elegans

*C. elegans,* where both *GULO* and *Regucalcin* (Regucalcin precedes GULO in the animal VC synthesis pathway) genes are missing, synthesizes VC [115]. Indeed, after exposing individuals to ^13^C-labeled *Escherichia coli* as food source, ^13^C was found in the VC pool, which led to the conclusion that VC must be synthesized de novo [115]. It was also observed that total VC content did not correlate with the concentration of sugar precursors commonly used by animals or plants, as well as their relevant derivatives [115]. Nevertheless, it is not clear that an increase in such sugar precursors would necessarily lead to an increase in VC synthesis [115]. In addition, a comprehensive search for homologs of the enzymatic constituents of the plant and animal VC synthesis pathways in *C. elegans* was also performed, revealing that no pathway was present as a complete set [115]. The loss of the *GULO* and *Regucalcin*, is also observed in other nematode species [118]. Given the evidence, it was suggested that VC synthesis in *C. elegans* likely occurs through a yet undescribed metabolic pathway [115]. No enzymatic constituents have been proposed for this pathway [115].

### 3.2. VC Synthesis in Drosophila Melanogaster

In insects, the need for VC synthesis, or adequate overall VC levels, varies depending on the species group that is considered [119]. In species from the Diptera, Lepidoptera and Coleoptera taxonomic groups, VC plays an important metabolic role [119]. Indeed, VC synthesis has been observed in the dipteran *D. melanogaster* [120] and in the lepidopteran *Bombyx mori* [116]. *Massie* et al. [120] were the first to note that *D. melanogaster* Oregon-R and Swedish-C male flies maintained for over 10 generations in food and water sources deprived of VC, displayed measurable total body VC concentrations. Moreover, by observing the continuous decrease in VC content in flies with age, these authors suggested that VC synthesis happened during the developmental stages, rather than in adult individuals [120]. Nevertheless, the amount of total VC increases in adult flies after exposure to a brief cold shock [120], indicating that in *D. melanogaster* VC synthesis might be also involved in stress response mechanisms. The authors evaluated the impacts of supplementation with L-Gulonolactone, D-Galactonolactone, and Glucose in adult flies, but none of these substrates contributed to increased VC content after the cold shock stimulus [120]. As the authors discussed, the enzymatic mechanisms might simply not be stimulated by dietary precursors [120]. This would suggest that *D. melanogaster* is able to synthesize VC precursors above the limiting rate of consumption, and that this self-sufficiency overshadows their addition. The authors, however, did not address the role of the flies’ gut microbiome in the synthesis process. This assessment is essential to determine the source of the VC found in the whole body homogenates, as several microbes have been found to synthesize and supply vitamins to hosts incapable of endogenous synthesis [121], and VC synthesis by insect symbionts has been previously observed [122].

Based on these findings, *Henriques* et al. [114] have demonstrated that axenic *D. melanogaster* Oregon-R flies possess similar levels of VC relative to normal individuals, and that the ex vivo culture of the microbiome did not lead to any VC accumulation in the resulting bacterial pellet and supernatant. Thus, Oregon-R flies’ microbiome does not significantly contribute to VC synthesis [114]. The impact of flies’ gender and sexual activity, not considered in [120], was also evaluated [114]. VC content in virgin flies had no significant difference relative to mated flies, but females had approximately three times more VC than males, and this observation could not be exclusively explained by the different bodyweight between genders [114]. This result implies that female flies can demonstrate increased amounts of VC due to: (i) the overexpression of the VC synthesis pathway; (ii) the presence of gender-specific structures, such as the ovaries; and (iii) increased VC storage and retention. Plasma VC concentrations are also higher in human and mouse females [123,124], but by a slight margin when in comparison to what is observed in *D. melanogaster*. The VC difference observed in mice may be explained in part by distinct renal VC excretion rates between males and females, since females are able to retain more VC from urine than males [125]. Therefore, gender-specific VC excretion rates along the *D. melanogaster* Malpighian tubule system, as well as the 50% difference in bodyweight between the sexes [114], might account for the observed results. These hypotheses, however, remain to be experimentally addressed.

It has been suggested that the physiological responses of insects can vary considerably depending on the cold exposure conditions [126]. Thus, as a complement to the brief cold shock exposure experiment reported in [120], *Henriques* et al. [114] verified the effects of extended cold exposure on the VC content of *D. melanogaster* Oregon-R. Unlike what was observed in [120], flies acclimated one day at 15 °C had a 23% decrease in VC levels relative to flies kept at 25 °C [114]. Nevertheless, cold-acclimated flies were able to replenish base VC levels after one day of recovery at 25 °C [114]. Therefore, different cold exposure conditions impose contrasting changes in *D. melanogaster*’s VC metabolism. Interestingly, the gene that encodes the lactonase that precedes GULO in the animal VC synthesis pathway, *Regucalcin*, has a duplicate in Sophophora species such as *D. melanogaster*, called *Drosophila cold acclimation* (*Dca*) [127,128]. The functions of this gene are still being determined, but so far they appear to be correlated with the response to cold exposure and wing size variation [127,128,129,130]. In regard to cold exposure, *Dca* has contrasting expression changes depending on the applied stress, since it is upregulated under acclimation conditions [126,128], but downregulated under cold shock [131,132]. This information implies that *Dca* expression changes are likely not correlated with the VC synthesis changes that occur under cold exposure responses in Sophophora species. Nevertheless, without further experimental data, the involvement of Dca in VC synthesis under cold exposure conditions cannot be yet ruled out. A direct involvement of Dca or Regucalcin in VC synthesis would imply that at least the lactonase component of the animal pathway is conserved in *D. melanogaster*. Using an in silico methodology, both *Regucalcin* and *Dca* were inferred to be involved in calcium homeostasis and in the oxidative stress response, but so far, no evidence was found for its involvement in VC synthesis [118]. The involvement in the calcium homeostasis and in the oxidative stress response are expected given the multiple functions of *Regucalcin* in vertebrates (reviewed in [133]). The inference of a divergent protein interaction surface at Dca in comparison with Regucalcin, along with the detection of positively selected amino acids near and within the active site region, suggests that Dca might have also acquired novel functions involving its lactonase activity and overall contribution to protein complexes [118]. The constitutive suppression, under the actin 5C driver, of either *Regucalcin* or *Dca* in *D. melanogaster,* did not clarify the role of these genes in VC synthesis [118]. Indeed, both genes are required for *D. melanogaster* viability, since in both cases over 70% of the individuals with suppressed expression were only able to reach the pupal stage [118], and thus the biological processes in which they are involved could not be determined. The reduced viability could be attributed, for instance, to the dysregulation of calcium homeostasis/signaling, as stated in [118]. Nevertheless, a cascade of metabolic events caused by VC deprivation could also explain the negative effects, especially when considering the previous evidence that VC synthesis occurs mainly during the developmental stages [120], which coincides with the lethal phase observed in the fly RNAi experiments [118]. In *B. mori*, a *Regucalcin*-like gene has been inferred to be involved in VC synthesis [116] (see below).

### 3.3. VC Synthesis in B. mori

In the lepidopteran *B. mori,* where *GULO* is absent, an alternative pathway for VC synthesis has been proposed, based upon the identification of a protein with GULO-like activity [116]. Evidence for VC synthesis comes from the observation that during silkworm development, the concentration of this nutrient is stable or even increased during stages without an active consumption of VC from mulberry leaves [116]. As in *D. melanogaster* [114], the authors addressed the impact of the microbiome, and found that it does not significantly contribute to VC synthesis [116]. Furthermore, the contribution of VC regeneration mechanisms in VC content fluctuations was also dismissed, since DHA levels remained unaltered during egg development [116]. Thus, it was hypothesised that *B. mori* is capable of endogenous VC synthesis [116]. Using a bioinformatics approach, *Hou* et al. [116] were able to identify two candidate GULO-like sequences (BGIBMGA012624 and BGIBMGA005735). Because BGIBMGA005735 codes for a protein that has a domain with similarity with another aldonolactone oxidoreductase (AO), it was selected for further analyses [116]. Enzymatic activity assays showed that this GULO-like protein is capable of VC synthesis using the same substrate as GULO, and that its activity is in part modulated by physiological changes that occur in the fat body during pupae development, implicating this tissue in the process [116]. This information is intriguing, as the *B. mori* fat body is considered to be analogous to the vertebrate liver and adipose tissue [134], and VC synthesis is known to occur in the liver of higher vertebrates [109,135]. Nevertheless, it is believed that in vertebrates, VC synthesis transitioned from the kidney to the liver during evolution and not the opposite [109], as now suggested by the observation that in protostomians the tissue responsible for VC synthesis is the structure that is analogous to the liver. It is, however, conceivable that VC synthesis might have transitioned from the kidney analogous tissue to the fat body in insects, in a manner similar, but independent of what is observed in vertebrates. In *B. mori*, the siRNA suppression of the *GULO*-*like* gene expression in the pupae fat body decreased VC synthesis but was ultimately non-lethal, while in the egg stage it considerably reduced overall survivability and led to delayed hatching [116]. Thus, although not ubiquitously essential for the complete growth of silkworms, this gene appears to have relevant functions implicated in viable transitioning from the egg to the larval stage [116]. The direct correlation between the reduced fitness of silkworm eggs and VC levels is, however, missing in that study. The authors note, however, that VC synthesis associated with the activity of the GULO-like protein studied, is not sufficient to fulfil the nutritional needs of developing individuals without additional supplementation [116]. The *B. mori* GULO-like protein is the ortholog of the vertebrate Delta(24)-sterol reductase (DHCR24) [136,137]. Like the vertebrate GULO, sterol reductases belong to the vanillyl alcohol oxidase/para-cresol methylhydroxylase (VAO/PCMH) flavoprotein family, but are usually known to catalyze distinct reactions [138]. While the DHCR24 protein is known to catalyze the NADPH-dependent reduction of a carbon–carbon double bond during the synthesis of cholesterol [138], using as substrate desmosterol that derives from endogenous [139] or dietary [140] sources, GULO is known to catalyze the oxidation of L-Gulonolactone, using O_2_ as an electron receptor, in the synthesis of VC [138,141]. In agreement with the functional divergence, phylogenetic studies also show that sterol reductases and AOs have diverged early in evolution [138]. Thus, it was not expected that a sterol reductase could metabolically replace GULO.

Considering that *B. mori* depends exclusively on mulberry leaves for nutrition purposes [142,143], and the VC content in these tends to differ depending on factors such as the climate or plant variety [144], the novel DHCR24 GULO-like functions might be the result of selective pressures to enhance VC availability during the development stages of this specific organism. Assuming that the reduced viability of silkworm eggs was caused by lower VC levels after *DHCR24* knockout, this evolutionary perspective could explain why VC synthesis mediated by DHCR24 seems to play an important but not essential role in silkworm growth and fitness. This interpretation correlates well with previous data that imply that some enzymes involved in insect sterol metabolism have likely evolved additional functions throughout evolution [145]. Nevertheless, the available data do not rule out the diminished sterol reductase activity as the partial or complete cause of the increased egg lethality.

Hou et al. [116] proposed that *B. mori* synthesizes VC through an adapted animal pathway in which the identified GULO-like enzyme functionally replaced GULO in the catalysis of the final oxidation step. This hypothesis appears to be the most parsimonious, since although GULO was lost in *B. mori*, the genes that encode enzymes that catalyze upstream enzymatic reactions in the animal pathway, such as *Regucalcin*, are present in this organism [118]. It should be noted, however, that this GULO-like enzyme is not present in *D. melanogaster* (by performing a “blastp” search (https://blast.ncbi.nlm.nih.gov/Blast.cgi, accessed on 28 July 2022) using the *B. mori* DHCR24 sequence (XP_004926865.1) as query, provided no homologous sequence hits in *D. melanogaster)* and thus, the *B. mori* VC pathway evolved independently from that of *D. melanogaster*. The nomenclature used in *Hou* et al. [116] can be confusing, since the authors refer to this animal modified pathway as the L-Gulose pathway, and state that it is analogous to the one found in plants. The proposed plant L-Gulose pathway has partial similarity with the animal D-Glucose pathway, since they both rely on L-Gulonate as intermediate substrate [146]. They differ, however, in the molecular mechanisms that lead to this step, as in the characterized animal pathway this molecule derives from D-Glucuronate and in plants from L-Gulose [146,147]. The metabolism of L-Gulose seems to be restricted to plants and archaea species [148], since the formation of L-Gulose likely involves a reaction catalyzed by a GDP-Mannose 3′,5′-Epimerase (GME) [107,148,149,150], and such epimerase activity was only found associated with the conversion of GDP-D-Mannose to GDP-L-Galactose in animals [151]. Considering these observations, and that naturally occurring L-Gulose is considerably rare [152], VC synthesis in *B. mori* likely does not rely on this precursor, and thus the nomenclature referred to in [116] should be interpreted with caution. Nevertheless, since plant GME can convert GDP-D-mannose to a mixture of GDP-L-Galactose, GDP-L-Gulose and GDP-D-Altrose [150], the contribution of enzymes with GDP-Mannose 3′,5′-Epimerase activity to protostomian L-Gulose metabolism, such as the GDP-4-keto-6-deoxy-D-mannose 3,5-epimerase/4-reductase of *D. melanogaster* [153] and its homologs in other species, cannot at present be excluded as playing a role in VC synthesis in protostomian species that do not use the described animal synthesis pathway.

### 3.4. The Role of Myo-Inositol in Animal VC Synthesis

Like in plants, the role of myo-Inositol as a precursor for animal VC synthesis through a partially alternative pathway has been addressed. The animal myo-Inositol Oxygenase (MIOX) converts this sugar to D-Glucuronate [154,155], which is a key molecule in the glucuronate-xylulose pathway [156,157] and an intermediate precursor in the vertebrate VC synthesis process ([17]; Figure 1). Experiments in *Cavia porcellus* (guinea pig) first suggested that myo-Inositol played an important role in VC synthesis or sparing [158], but later studies showed that this was not the case [159]. Indeed, there is no functional *GULO* gene in *C. porcellus*, and this is why this species is highly regarded as a scorbutic model organism when missing a dietary source of this nutrient [53,160,161]. The auxotroph nature of the guinea pig should have been considered a limiting methodologic factor in addressing the role of myo-Inositol in animal VC synthesis, as the absence of a functional VC synthesis pathway could invalidate the study of potential precursors in this organism. This criticism stands even considering that this polyalcohol is naturally present in the guinea pig [162,163] in sufficient concentration for viable VC synthesis [117]. The conclusions referred in [159] were thus, expected. In this sense, teleost species are also scurvy prone in the absence of VC supplementation [164,165], and thus should be avoided as experimental subjects in further explorations on the subject.

Myo-Inositol was found not to be involved in VC synthesis in *Rattus norvegicus* (rat) [166]. Unlike the guinea pig, rats are capable of synthesizing VC using the characterized animal pathway [167]. In placental mammal species capable of doing so, VC synthesis occurs in the liver since it relies on the hepatocyte glycogen reserves [168], and GULO is exclusively found in this tissue [109,135,169]. The D-Glucuronate needed for the process can be obtained as the by-product of the glucuronidation of endobiotic and xenobiotic compounds in the mammals liver [104,170,171], as well as the conversion of UDP-D-Glucuronate to D-Glucuronate by a putative UDP-glucuronidase [104] (see Figure 1). However, myo-Inositol is only metabolized in the vertebrate kidney [172], and, as expected given this observation, MIOX appears to be a kidney-specific enzyme [155,173]. The D-Glucuronate originated by the action of MIOX is likely mostly used in the glucuronate-xylulose pathway that is particularly active in the vertebrate kidney [174]. Therefore, although it is conceivable that myo-Inositol could play a role both in VC synthesis and in the glucuronate-xylulose pathways [175,176], it is likely that only a small amount of D-Glucuronate diffuses from the kidney to the liver, thus playing a negligible contribution in the context of VC metabolism in mammals that synthesise VC in the liver.

The observations made in rats may not apply to all mammal species capable of endogenous VC synthesis. Indeed, in egg-laying mammals (Monotremata) GULO is exclusively found in the kidney [177]. In the Peramelina order (Marsupials) GULO is found in both the liver and kidney, although in species from the Diprotodonta order (Marsupials), GULO is mainly found in the liver [177]. Therefore, in basal mammal groups, besides D-Glucose, myo-Inositol may also be a relevant substrate for VC synthesis in species where GULO is found in the kidney.

Hänninen et al. [117] compared the capacity for VC synthesis from myo-Inositol in the Leghorn hen (*Gallus domesticus*) kidney and rat liver. In [117], the findings of [166] were replicated, as rat liver extracts displayed no MIOX activity and were unable to synthesize VC from myo-Inositol [117]. However, the analyses performed on the galliform species capable of VC synthesis in the kidney [178] showed opposite results [117]. The authors were able to determine that ^14^C-labelled myo-Inositol was incorporated into the hen kidney VC pool, in a manner partly dependent on the MIOX ability to convert this sugar to D-Glucuronate [117]. In addition, the results also showed that the concentration of this polyalcohol in hens is sufficient for biologically relevant VC synthesis [117]. Not all birds have, however, GULO in the kidney, an essential requisite for VC synthesis in this organ. For instance, Passeriformes species synthesize VC in the liver [117].

In amphibians [179,180], reptiles [179,180] and basal fish groups [179,181], GULO is only found in the kidney as well. Therefore, although generally disregarded, the contribution of myo-Inositol to VC synthesis in animals, particularly in ancestral vertebrate species, cannot be excluded.

### 3.5. Summary and Future Perspectives

In this section we put forward hypotheses regarding VC synthesis pathways in nematodes and insects (Figure 2), and present possible methodological approaches for further exploration of this subject. The available information strongly suggests that VC synthesis in *C. elegans* occurs by a very distinct pathway from the one used by most animals, since it may not rely on the known substrates of the described pathways as precursors [115], and the two key enzymes in the process, namely Regucalcin and GULO [115,118], are also missing (see detailed discussion above). The precursor supplementation assays have some experimental limitations [115], that can be overcome using radiolabeled compounds, since this methodology allows the direct visualization of substrate incorporation and excretion rates [182]. In addition to the classical substrates, the assays could also be expanded to include myo-Inositol, since the vertebrate *MIOX* gene has an ortholog in *C. elegans* (*C54E4.5*; Ensembl accession number WBGene00016920), and the D-Glucuronate that is generated could contribute to VC synthesis as it happens in birds (Figure 2). The use of a complete or partial knockout of *C54E4.5* could also be used to address the involvement of this gene in VC synthesis by looking at VC levels that can be easily measured using the methodology described in [114].

The report of an AO function for the *B. mori DHCR24* sterol reductase [116] also raises questions regarding possible adaptations of non-AO VAO/PCMH flavoproteins for VC synthesis (Figure 2). *C. elegans* contains a functional *DHCR24* ortholog [183,184] that should be also studied in detail.

As in *C. elegans*, in flies, the supplementation of food with precursors naturally used in the animal VC synthesis pathway did not contribute to increased VC content [101] (see detailed discussion above). However, as suggested for *C. elegans*, the use of radiolabeled substrates is likely more adequate to address this issue. Such approach could be particularly relevant in deciphering the role of myo-Inositol in *D. melanogaster* VC metabolism, since a *MIOX* ortholog has been identified in this organism (*CG6910*; [185]; Figure 2). In *D. melanogaster,* under cold acclimation conditions, the expression of the *CG6910* gene is reduced [126]. *Dca* is overexpressed under cold acclimation conditions [126], but VC levels decrease during this stimulus [114], which may be due to the suppression of *CG6910* expression during the cold stress response.

Both Dca and Regucalcin likely catalyze the conversion of L-Gulonate to L-Gulono-1,4-lactone in *D. melanogaster*, and as such, if Dca is involved in VC synthesis, through a pathway resembling the one found in vertebrates, then the availability of its substrate might be a limiting factor for VC synthesis during cold acclimation conditions. L-Gulonate is obtained by the transformation of D-Glucoronate by the action of the aldo-keto reductases AKR1A1 and AKR1B1 in mammals [186,187] (Figure 1). The catalytical contribution of the two enzymes is very distinct, as AKR1A1 and AKR1B1 are responsible for approximately 85% and 15% of the conversion ratio, respectively [187]. The reduced contribution of AKR1B1 can be explained by its reduced specific activity rather than substrate specificity [188], reflected in the reduced efficiency for D-Glucuronate reduction when compared to AKR1A1 [187]. The human *AKR1A1* gene has no orthologs in *D. melanogaster* (Ensembl accession number ENSG00000117448), but the *AKR1B1* gene displays a one-to-many ortholog relationship (Ensembl accession number ENSG00000117448). One of these homologs, namely *CG10863*, encodes an enzyme known to share catalytic similarities to AKR1A1 and AKR1B1 [189]. This gene is under-expressed during cold acclimation, and D-Glucoronate is notably more abundant in cold acclimated flies [126]. Therefore, we hypothesize that under the mentioned stress conditions, VC synthesis might be impaired by the reduced expression levels of both *CG6910* (see above) and *CG10863*, the latter leading to the accumulation of D-Glucoronate in flies. Since, in contrast to *Dca,* that is overexpressed, *Regucalcin* is also under-expressed under cold acclimation conditions [126], when flies are placed in recovery conditions, along with the restoration of normal *CG6910* and *CG10863* expression levels, the role of Dca might be important for the rapid increase in VC synthesis, by providing a boost to the conversion of L-Gulonate to L-Gulono-1,4-lactone. Nevertheless, since Regucalcin and Dca are likely to perform important roles in calcium homeostasis as well [118], they may play also other important roles during cold exposure recovery. The regulation of calcium levels might be important for the maintenance of viable muscle function [126] in a manner unrelated to *D. melanogaster* VC synthesis. Thus, even if Dca does not intervene in VC metabolism, the reduced expression of *CG10863* and *CG6910* and *Regucalcin* under cold acclimation, and subsequent normalized expression on recovery conditions might be sufficient to explain the results obtained in [114]. While further studies on these genes may give insight into the VC synthesis pathway, there is still no candidate for the functional replacement of the *GULO* gene that has been lost in insects, such as *D. melanogaster*. Indeed, this species is a cholesterol auxotroph [190,191], that has lost the ability to convert desmosterol to cholesterol [192], likely due to the loss of the *DHCR24* orthologous gene (the one involved in VC synthesis in *B. mori* [116]).

## 4. Prokaryote AOs: An Unexplored Concept

### 4.1. The Role of AOs in VC Synthesis Evolution

The divergent VC synthesis mechanisms here discussed suggest that the majority of species rely on AOs to catalyze the final metabolic step. The distribution pattern of AOs in eukaryotes is complex (Figure 3), and the evolutionary history that led to this scenario is still lacking definition. A very comprehensive study has previously explored the emergence of distinct VC synthesis pathways based upon the eukaryote AOs dispersion, while evaluating the contribution of putative lateral gene transfer events in the process [94]. Although the evolutionary origin of the plant *GALDH* could not be inferred, it was hypothesized that *GULO* (used in this context as the representative of both *GULO* and *ALO*) and *GALDH* could be the result of an AO gene duplication event that occurred in the ancestral eukaryote (Figure 3) [94]. Alternatively, it was also proposed that *GALDH* might have arisen in the Archaeplastida ancestral (Figure 3), and ultimately replaced *GULO* in many non-glaucophyte lineages due to the increased oxidative stress imposed by the primary endosymbiosis event that led to chloroplast emergence [94]. The authors concluded that this was the most parsimonious explanation for *GALDH* emergence, considering that VC synthesis catalyzed by GALDH does not contribute to ROS production, and thus increases the availability of anti-oxidant mechanisms to deal with ROS originated by chloroplast metabolism [94]. Then, possible secondary endosymbiotic gene transfer (EGT) events could have been responsible for the transfer of *GALDH* to the SAR and Euglenida lineages, while horizontal gene transfer (HGT) events were the likeliest cause of *GALDH* in the opisthokont Choanoflagellata [94]. Although a putative *GALDH* was uncovered in the Chytridiomycota fungi *Gonapodya prolifera* [94], there was no discussion on the acquisition mechanism responsible for the presence of this AO in this taxonomic lineage. Nevertheless, considering the proposed hypothesis, this might also be the outcome of a putative HGT event, as the Chytridiomycota are aquatic species inferred to have interacted with streptophyte algae throughout evolution [193].

Both hypotheses consider that the ancestral eukaryote had at least one functional AO to catalyze the last step of VC synthesis. This assumption raises additional questions regarding the emergence of AOs, as it does not clarify if they have prokaryotic origins or are the result of the evolution of de novo genes that arose in the ancestral eukaryote. In an extreme scenario, it could be proposed that *GULO*, *GALDH*, or even both genes, did not exist in the eukaryote ancestor, and were later on acquired through EGT from the alphaproteobacteria and cyanobacteria that are thought to have led to the emergence of mitochondria and chloroplasts, respectively [197,198]. This is a viable hypothesis, considering that some of the genomic information of these organelles has been transferred to nuclear DNA throughout evolution [199,200,201].

The possibility of endogenous VC synthesis in bacteria is often disregarded by many scientific publications on the subject, and this may be the reason why the acquisition of AO genes in the ancestral eukaryote by EGT is not considered as a current alternative scenario. However, although relatively scarce, evidence for prokaryote VC synthesis has been published before [87,88,89,90,91,92,93,122]. In some cases, the authors were able to characterize L-Gulono-1,4-lactone Dehydrogenases (GUDH) as the likely AOs that catalyze the final step of synthesis of this nutrient in bacterial species [87,90]. These GUDH have high substrate specificity to L-Gulono-1,4-lactone and do not metabolize L-Galactono-1,4-lactone [202,203], and therefore have more functional similarities to the animal GULO and fungi ALO proteins than to plant GALDH. However, they are unable to use oxygen as an electron acceptor, but can use cytochrome C to this effect [203], bearing some similarities to the plant GALDH. Based on the hybrid functional characteristics of bacterial GUDH, if such an AO of prokaryote origin is related to the emergence of VC synthesis in the ancestral eukaryote, then the *GULO* and *GALDH* lineages could be the end result of adaptation to the cell conditions present in the distinct taxonomic groups. This hypothesis implies a single gene origin for the various AO lineages observed in eukaryotes and is compatible with the ancient paralogy scenario presented in [94], under the assumption that this AO was duplicated before the emergence of the main eukaryote lineages. However, *Arabidopsis thaliana* has a functional GUDH enzyme with similar characteristics to the GUDH found in *Mycobacterium tuberculosis* [202,203], thus implying that some Streptophyta can have both GUDH and GALDH. This could mean that the *GALDH* lineage might have derived from a duplication of *GUDH* in the Archaeplastida after the divergence from glaucophytes (Figure 4A), thus reinforcing the role of LGT in the evolution of AO [94]. Nevertheless, if *GALDH* and *GULO* derived from a *GUDH* duplication in the ancestral eukaryote, the identified *A. thaliana GUDH* could be a remnant of an ancestral prokaryote gene (Figure 4A), an hypothesis that correlates well with its phylogenetic outgroup positioning relative to the *GALDH* and *GULO* gene clusters in previous phylogenetic analyses [94,203].

If AOs had a prokaryotic origin, bacteria to eukaryote HGTs might also have played a role in the emergence of some VC synthesis mechanisms, since HGT events are known to have been important in the evolutionary adaptation of eukaryotes [204,205]. Such events appear to be widespread, as evidence for such phenomenon has been reported in protist [206], plant [207], fungi [208] and animal [209] species. HGT events seem to be more common in fungi than in animals and plants [210], and in some cases, fungi appear to have acquired entire metabolic pathways [211,212]. In the context of VC metabolism, several fungi species are able to synthesize VC analogues, such as D-erythroascorbate, through the two-step conversion of D-arabinose, L-fucose, L-xylose or L-galactose [102,213]. Interestingly, older reports have suggested that bacterial species are also capable of synthesizing a molecule with the antiscorbutic properties of VC from xylose, although without conclusive identification of the compound produced at the time [92,93]. Bacteria are known to have very few introns across their genome [214], and the fungi *S. cerevisiae ALO* gene is encoded by a single exon (EnsemblFungi accession number YML086C). Nevertheless, this species is known to have a particularly low intron presence across the genome when compared to other species [215,216], and thus such information cannot be used as evidence of HGT. However, other examples are available, such as the single exon *ALO* gene of *Scheffersomyces stipites* (EnsemblFungi accession number PICST_88335) and *Aspergillus nidulans* (EnsemblFungi accession number ANIA_00836). These fungi sequences might represent remnants of assimilated prokaryotic DNA. Considering the information presented, it would not be implausible to assume that the fungal D-erythroascorbate biosynthetic pathway could be related with the complete or partial acquisition and conservation of a pre-existent bacterial pathway in the Opisthokonta ancestral, which later on diversified after speciation (Figure 4B). To our knowledge, no study was yet performed to evaluate the presence and distribution of AOs in prokaryotes, and since phylogenetic methods can help track HGT events [217], we find that there is still much more to explore in this respect.

Although speculative, this discussion clearly highlights the limitations imposed by the exclusion of prokaryotes from VC synthesis studies. As a preliminary step, the inference of putative AOs in such basal lineages could provide the foundation for further findings on this subject.

### 4.2. AOs Presence in Archaea and Bacteria

In [94], the authors state that an extensive homology search led to the inference of only one significant *GULO* homolog in prokaryote species. The initial search was performed using the *M. musculus* GULO and *A. thaliana* GALDH sequences as query in a “blastp” approach [94]. Nevertheless, since *ALO* sequences represent a distinct lineage within the *GULO* clade [94], the use of a representative ALO protein sequence as query can still improve the likelihood of encountering additional homology hits. Moreover, the rapid increase in genome availability for uncultured bacteria [218], as well as the use of more sensitive methods to detect remote (distant) homologs than BLASTp, may help fill some gaps in prokaryote AOs distribution patterns. Such data are needed to confidently address the HGT hypotheses discussed above, and should the subject of future research. Nevertheless, a “blastp” search, coupled with Bayesian phylogenetic inferences, as here performed, already provides some interesting clues, despite the low posterior credibility values of some branches (Figure 5).

One of the most notorious results is the well-supported close relationship between sequences from the cyanobacteria *Moorena sp. SIO4G3* and *Calothrix parasitica NIES-267* and the parasitic protists *Trypanosoma brucei brucei TREU927* and *Leishmania major* strain *Friedlin*. The symbiotic relations of these protists with bacteria provide advantageous nutritional benefits and promote HGT events [225]. Indeed, it has been argued that prokaryote to eukaryote HGTs of genes related to vitamin synthesis and salvage mechanisms are likely to have occurred in Trypanosomatidae species [225]. The observation made here seems to follow the same trend. A previous report indicated that *Trypanosoma cruzi* and *T*. *brucei* possess a L-Galactono-1,4-lactone oxidase (GAL) capable of synthesizing VC using the primary plant and fungi substrates, namely L-Galactono-1,4-lactone and D-Arabinono-1,4-lactone, but with negligible affinity to the known vertebrate substrate, L-Gulono-1,4-lactone [226,227]. Furthermore, one species of the same genus of *L. major*, *Leishmania donovani*, has an ALO that is also capable of VC synthesis using D-Arabinono-1,4-lactone as substrate, but not L-Gulono-1,4-lactone [228]. Although not yet shown, given the close phylogenetic relationship relative to *Trypanosoma* sequences, it is possible that the *L. donovani* ALO has affinity to L-Galactono-1,4-lactone as well. Therefore, the *Moorena sp. SIO4G3* and *C. parasitica NIES-267* sequences may represent genes capable of encoding AOs that are able to synthesize VC using L-Galactono 1,4-lactone and D-Arabinono-1,4-lactone, but not L-Gulono-1,4-lactone. The close phylogenetic relationship between the sequences of *Asgard group archaeon* and several fungi *ALO*, as well as the considerable sequence clustering of the majority of bacteria, one fungi and one Archaea with the *M. tuberculosis GUDH*, constitute additional indication of possible shared functional characteristics between the encoded enzymes. If so, this would suggest that some Archaea are capable of utilizing all of the common substrates for VC synthesis to a variable degree, while some Proteobacteria, Actinobacteria, Bacteroidetes, Archaea and fungi species could have specialized in the exclusive use of L-Gulono-1,4-lactone. The *M. tuberculosis GUDH* lineage is clearly divergent from the *A. thaliana GUDH,* revealing that although the biological properties are similar between them, they likely have distinct origins. The latter lineage does not appear to be restricted to *A. thaliana*, indicating that *GUDH* might be present and intervene in VC synthesis in more Viridiplantae and even fungi. The *A. thaliana GUDH* appears to have been duplicated many times, a result consistent with previous observations of a high number of homologous AO sequences in this species [229]. The phylogenetic relationships between the *ALO*, *GULO*, *GUDH* and *GALDH* lineages vary considerably relative to those found in [94]. However, the clustering patterns of the Bacteria and Archaea sequences relative to those lineages are very well supported, implying that they likely represent relevant AOs.

The currently described AOs are characterized in the literature by the presence of amino acid positions thought to be related to their biological functions [202,226,230,231,232]. Following on these criteria, a search for relevant protein regions was performed (Table 1) using the alignment file that was used for the phylogenetic inference here presented (Appendix A). To simplify the visualization, the total number of representative sequences was reduced in the results shown (Table 1). Despite this, the evidence still illustrates a convincing degree of amino acid conservation between putative bacteria and archaea AOs relative to the reference sequences presented. The putative prokaryote AOs that were identified are likely able to covalently bind to flavin adenine dinucleotide (FAD) and have efficient catalytic activity, since the essential histidine (H), tryptophan (W) and lysine (K) residues distributed across the N-terminal FAD-binding domain and HWXK motifs [202] are conserved. Furthermore, the Gluthamine (E)–Arginine (R) pair usually involved in the stabilization of the lower electric potential derived from flavin reduction and AO substrate binding and specificity [202] also appears to be conserved. With the exception of the *M. tuberculosis* GUDH [87], it is thought that the presence of a specific Glycine (G) residue within the amino acid sequence of AOs confers the ability to use oxygen as electron acceptor [202]. If so, the putative prokaryote AOs can act as oxidases, similarly to animal GULO and fungi ALO. Moreover, a cysteine (C) residue linked to substrate specificity of AOs [202] is present in many of the putative prokaryote AOs, with the exception of *Bacteroidetes bacterium*. In agreement with the phylogenetic results, the overall conservation of essential amino acid positions suggests that the putative prokaryote proteins here displayed represent novel AOs.

As previously stated, the analyses performed in this section were primarily focused on the verification of the existence of putative AOs in prokaryote species. Although there are no functional data to support these inferences, the results here presented raise doubts regarding the common assumption of generalized prokaryote VC auxotrophy, and provide a new perspective on the possible distribution of VC synthesis mechanisms in the different life kingdoms (Figure 6). The Asgard archaea lineage has been regarded as the prokaryote origin of the ancestral eukaryotes [233,234], and chloroplasts are organelles derived from cyanobacteria endosymbiosis [198]. In this sense, if functional AOs are present in species from both taxonomic groups as the in silico analyses suggest, VC synthesis evolutionary models must consider them.

The presence of AOs in prokaryotes may open new perspectives for potential industrial applications. VC synthesis usually relies on an initial chemical conversion of D-Glucose to D-Sorbitol followed by two independent fermentation steps using *Gluconobacter oxydans* and a mixed culture of *Ketogulonigenium vulgare* and *Bacillus megaterium*, respectively, to obtain 2-Keto-L-Gulonic acid [235], a VC precursor. This two-step fermentation method is highly efficient, but the hydrogenation of D-Glucose and the maintenance of balanced co-cultures still have economical and labor costs that can, in principle, be avoided [235], using for instance, a direct one-step fermentation process [235,236]. This can be achieved through the genetic integration of plant VC synthesis pathway components, such as the *A. thaliana GALDH*, in fungi [236], as well as the integration of genes involved in D-Sorbitol/L-Sorbose metabolism from different prokaryote sources into a single bacterial chassis [237,238,239]. Substantial optimization is, however, required since the efficiency of VC synthesis is still dependent on the adequate availability of cofactors and/or compatible electron acceptors [235,236]. Nevertheless, if the species used as chassis already has the majority of the required enzymatic components, such limitations could be overcome. Some γ-proteobacteria species, such as *Erwinia herbicola*, are able to perform the direct oxidation of D-Glucose to 2,5-diketo-D-Gluconic acid [240], and therefore constitute a suitable bacterial chassis candidate for 2-Keto-L-Gulonic acid synthesis with reduced genetic manipulation and kinetic optimization. This possibility has been successfully explored through the integration of an exogenous 2,5-diketo-D-Gluconic acid reductase into *E. herbicola*, that allows the production of 2-Keto-L-Gulonic acid from 2,5-diketo-D-Gluconic acid [240] or after the fusion of a *Corynebacterium sp.* protoplast with *E. herbicola* [241]. The effective yields were low in both cases, but such approaches still remain relevant for potential industrial applications [235]. The phylogenetic results here presented imply the presence of a putative AO in *P. aeruginosa*, which is also a γ-proteobacteria. Therefore, the metabolic components that are present in *E. herbicola*, might also be present in this species. If true, *P. aeruginosa* could represent an additional bacterial chassis that could be explored for the purpose of VC synthesis. Given that a bioengineered *P. aeruginosa* acid phosphatase has already been used to convert VC to a more stable L-Ascorbic acid-2-phosphate derivative resistant to high-temperatures [242], the same bacterial species could be used for both VC synthesis and stabilization, a relevant concern in the recovery process of this nutrient [236].

The current knowledge and hypotheses on novel VC synthesis pathways here discussed (summarized in Figure 6) are expected to promote research on this issue in different organisms.

## 5. Conclusions

Current research has only grasped a fraction of the VC contributions to overall organism function, as well as the complexity behind the endogenous synthesis of this nutrient. Here, we critically review the literature on metazoan VC synthesis and put forward novel hypotheses and discuss methodological approaches that may contribute to the characterization of alternative enzymatic mechanisms behind VC synthesis in *C. elegans* and *D. melanogaster*. A putative prokaryotic origin for eukaryote *AO*s is also hypothesized and discussed in the context of the current evolutionary scenario that led to the emergence of the *GULO* and *GALDH* lineages. The performed phylogenetic analysis indicated that putative AOs are present in bacteria and archaea species. These inferences are discussed in the context of possible applications on the optimization of the industrial production of VC.

## Figures and Tables

**Figure 1 genes-13-01917-f001:**
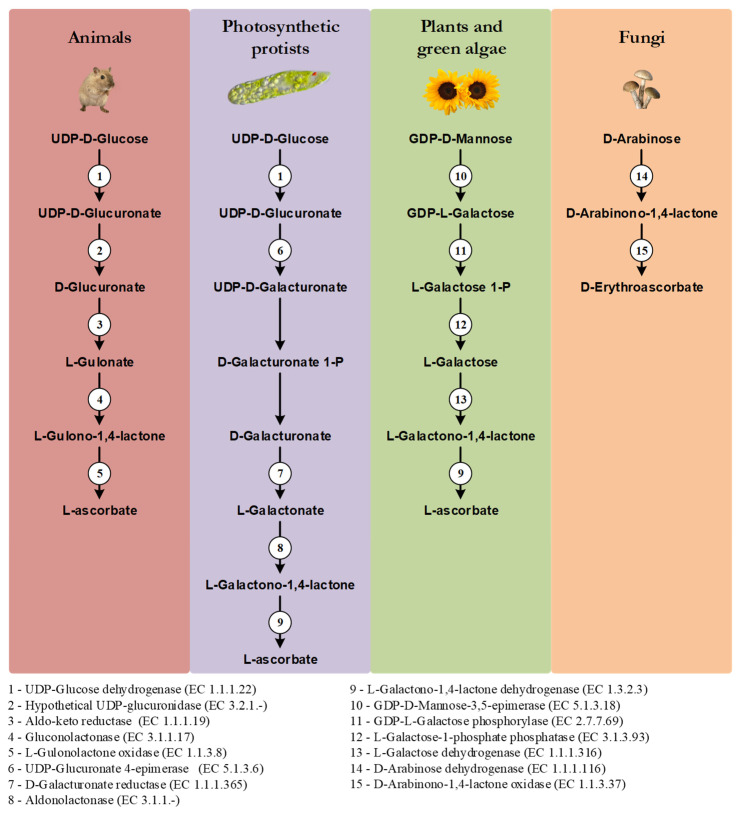
Graphical illustration of the main VC and D-erythroascorbate synthesis pathways, adapted from the MetaCyc database [108]. The numbered circles represent enzymes that are known to catalyze the substrate conversions within the pathways (see the enzyme list presented below the illustration). Reaction steps without an attributed enzyme are performed by unknown enzymes.

**Figure 2 genes-13-01917-f002:**
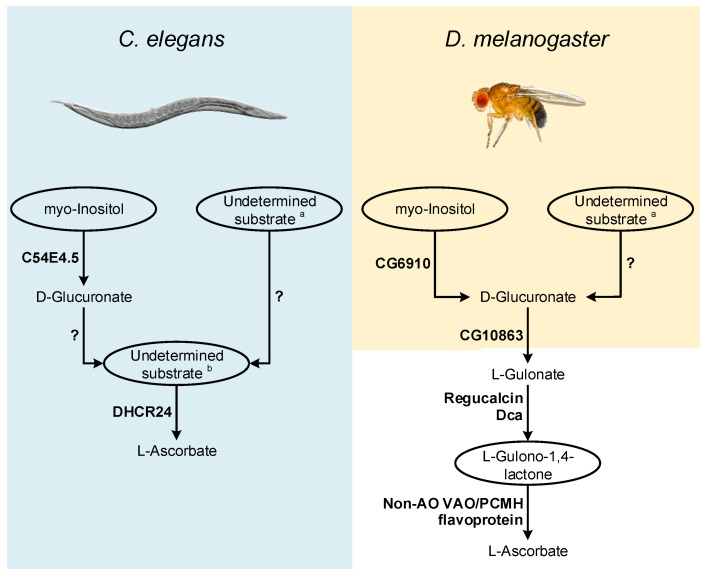
Inferences on enzymatic mechanisms and possible substrates utilized by *C. elegans* and *D. melanogaster* in the VC animal synthesis pathways. The ellipses highlight the main substrates possibly used in the proposed pathways, but that remain to be addressed using appropriate experimental setups. Enzymes that may be correlated with VC synthesis in these species are highlighted in bold next to the metabolic steps they are likely to catalyze. The interrogation marks indicate uncertainty in the steps regarding mechanisms of conversion of uncharacterized substrates, and the relative contribution of these processes to the availability of VC precursors. Common sugars (such as D-mannose, L-Galactose, D-Glucose and D-Arabinose) are assigned with ^a^ while L-Galactono-1,4-lactone, L-Gulono-1,4-lactone and D-Arabinono-1,4-lactone are assigned with ^b^.

**Figure 3 genes-13-01917-f003:**
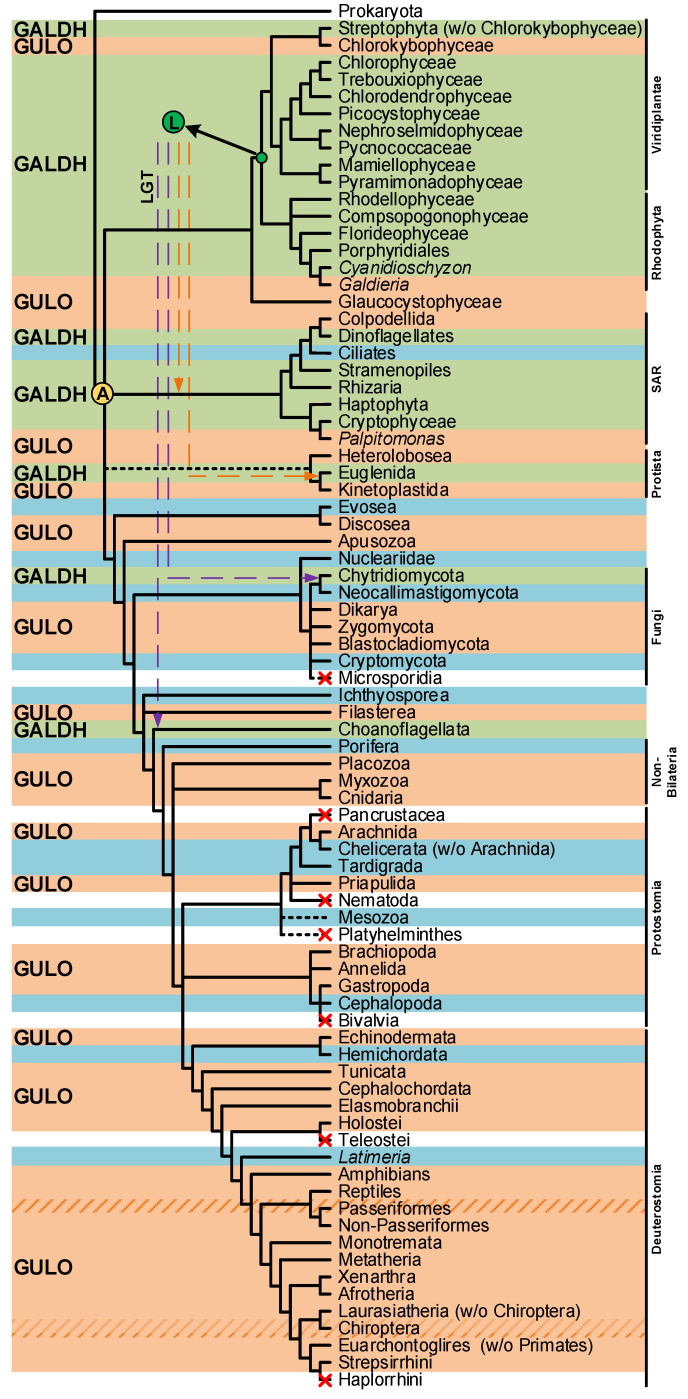
Presence/absence pattern of the *GULO*/*GALDH* lineages in eukaryotes. The taxonomic groups where the *GULO* lineage is present are highlighted by an orange background, while those where the *GALDH* lineage is present are highlighted by a green background. Independent *GULO* loss events within specific vertebrate groups are represented by stripe patterns. The blue background highlights uncertainty regarding either gene lineage presence/absence. These cases represent groups where both gene lineages could not be found in less than three species, or groups where the sequences identified were likely derived from genome contamination. Taxonomic groups where either gene lineage was lost are not highlighted, and a red cross can be seen in overlapping the branches they represent. Higher taxonomic ranks were added to the right of the figure to facilitate the interpretation. The ancestral paralogy and lateral gene transfer hypotheses proposed in [94] are indicated by the yellow and green filled circles with an A and L acronyms over the relevant tree nodes, respectively. The implied duplication events occurred during the time scale represented by the branches that led to the highlighted nodes. Both gene lineages already existed in scenario A, and the pattern of loss/presence of each was based upon selective constraints of the distinct taxonomic groups. Scenario L implies the origin of the *GALDH* lineage in the common ancestor of the Rhodophyta and Viridiplantae, that was later transferred to other taxonomic groups by endosymbiotic (orange dashed arrows) or horizontal (dashed purple arrows) gene transfer events. The data depicted in the figure was obtained from [94,112,114], and the taxonomic relationships are based on those found in [194,195] and the Tree of Life Web Project (http://tolweb.org/tree/; [196]; last accessed on 12 October 2022).

**Figure 4 genes-13-01917-f004:**
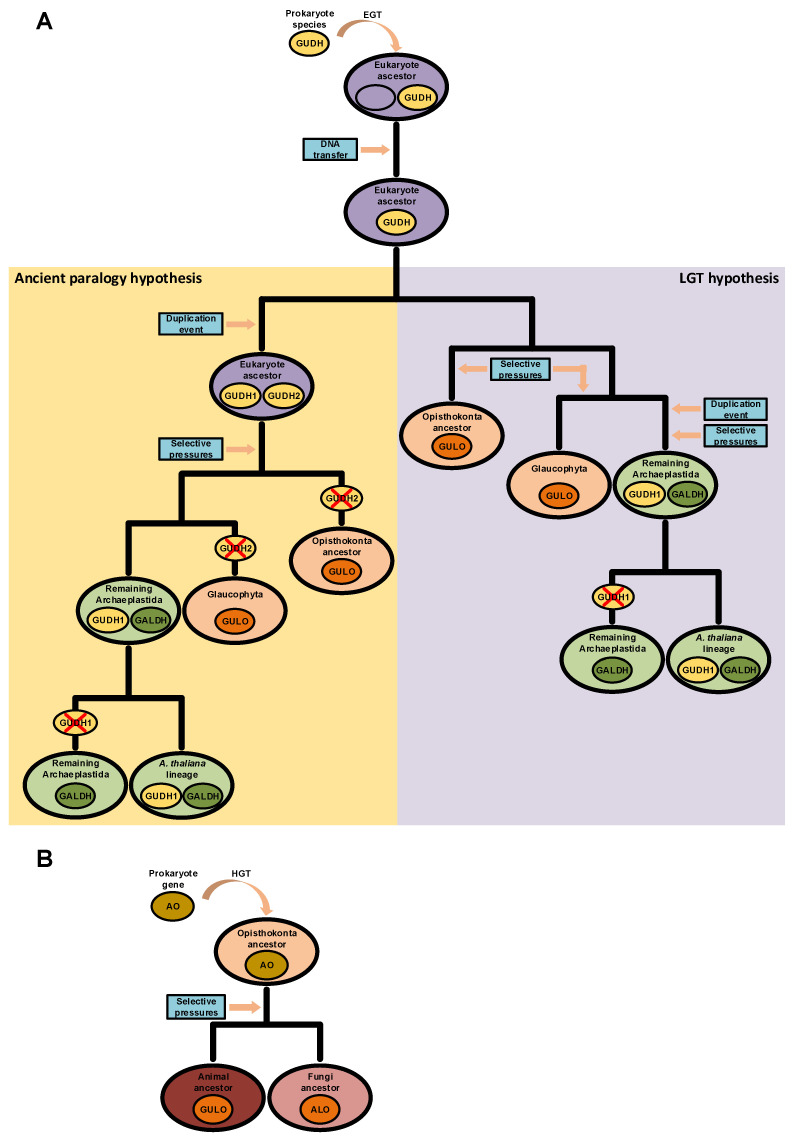
Putative scenarios of eukaryote AO evolution derived from prokaryote LGT events. In panel (**A**), a prokaryote *GUDH* was initially integrated in the genome of the ancestral eukaryote after an EGT event. Two hypotheses for the divergence of *GUDH* in eukaryotes are presented. In the left (yellow background), the *GUDH* present in the ancestral eukaryote was duplicated, originating *GUDH1* and *GUDH2*. Distinct selective pressures led to the loss of the *GALDH* ancestral gene (*GUDH2*) in glaucophytes and opisthokonts, and thus *GULO* is the result of *GUDH1* evolution. In the non-glaucophyta Archaeplastida, *GALDH* was the result of selective pressures imposed on *GUDH2*, while *GUDH1* maintained the ancestral state. During evolution, *GUDH1* was eventually lost in taxonomic groups that diverged from the one where *A. thaliana* is included. This scenario is compatible with the ancient paralogy hypothesis presented in [94], as indicated in the figure. In the right (purple background), the scenario represents the direct origin of the opisthokont and glaucophyte *GULO* lineages as the result of selective pressures imposed on the ancestral *GUDH*. In this case, a *GUDH* duplication event affected only the Archaeplastida ancestor after the divergence from the glaucophytes, and the *GUDH1* and *GALDH* duplicates are the result of distinct selective pressures. Then, *GUDH1* was lost in taxonomic groups that diverged from the one where *A. thaliana* is included. This scenario is more compatible with the LGT hypothesis presented in [94], as indicated in the figure. In panel (**B**), *GULO* and *ALO*, in animals and fungi, respectively, are the result of selective pressures imposed on an AO acquired through HGT by the ancestral opisthokont.

**Figure 5 genes-13-01917-f005:**
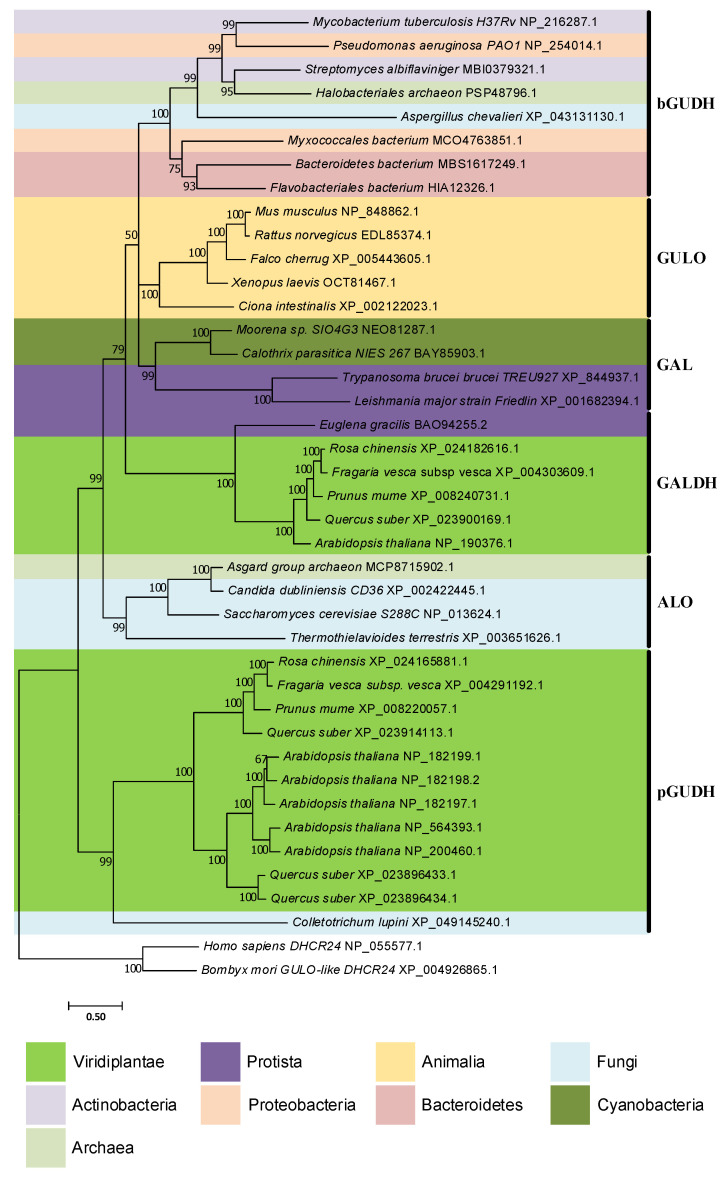
Bayesian phylogenetic analysis of eukaryote and putative prokaryote *AO*s. Two *DHCR24* sequences from *Homo sapiens* and *B. mori* were used to root the tree. The remaining sequences are colored according to the relevant taxonomic groups analyzed, following the legend available below the phylogeny. The likely type of AO represented in each inferred sequence cluster can be seen to the right of the phylogeny. The GUDH, GULO, GAL, GALDH and ALO acronyms are indicative of L-Gulono-1,4-lactone Dehydrogenase, L-Gulonolactone oxidase, L-Galactono-1,4-lactone oxidase, L-Galactono-1,4-lactone dehydrogenase and D-Arabinono-1,4-lactone oxidase, respectively. The bacterial and plant GUDHs are distinguished by the “b” and “p” prefixes, respectively. The sequences used for the phylogenetic inference were obtained using the “blastp” algorithm integrated in the NCBI database (https://blast.ncbi.nlm.nih.gov/Blast.cgi; accessed on 26 July 2022) and the *S. cerevisiae* S288C ALO sequence (NP_013624.1) as query against the genomes of Actinobacteria, Proteobacteria, Bacteroidetes, Cyanobacteria, Archaea, Viridiplantae, fungi and Euglenozoa species. This genome selection considered bacterial groups previously correlated with VC synthesis [87,88,89,90,91,92,93], as well as the complementation of prokaryote taxonomy by the inclusion of Archaea. The Viridiplantae, fungi and Euglenozoa groups were selected to provide methodological validation relative to current literature and overall phylogenetic context. Homologous sequences representative of two species of each bacterial group and Archaea, five species of Viridiplantae, five species of fungi and three species of Euglenozoa were extracted. The “blastp” output produced considerably more results for each taxonomic group than those extracted, but the restrictive selection is representative, and through the use of more elaborated methods that have limitations in relation to the number of sequences they can handle, the quality of the phylogenetic inferences was improved. In addition to these sequences, five animal *GULO* sequences identified in [112] were also included to facilitate the interpretation of the phylogenetic results, as well as two sterol reductases sequences encoding the human and *B. mori* DHCR24 proteins, to be used as outgroup. The SEDA software [219] was used to verify the datasets and remove any anomalous sequence or isoforms that could be present. The Bayesian phylogenetic inference was performed using a MUSCLE [220] alignment as implemented in T-Coffee [221] and 2,000,000 tree generations with a defined burn-in of 25% as parameters for MrBayes [222], within ADOPS [223]. For both SEDA and ADOPS, the Docker images that are available at the pegi3S Bioinformatics Docker Images Project (https://pegi3s.github.io/dockerfiles/; accessed on 26 July 2022; [224]) were used.

**Figure 6 genes-13-01917-f006:**
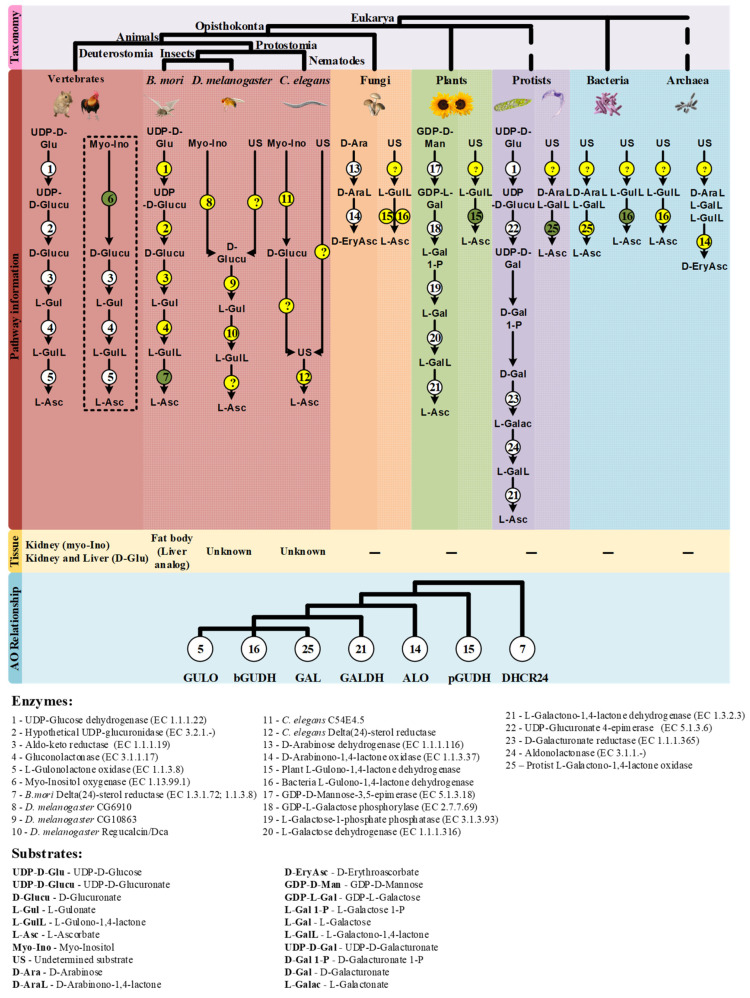
Summary of the findings discussed in this review. The taxonomic relationship between the species mentioned in this work is displayed on the top of the figure. The characterized VC synthesis pathways and the putative alternatives in eukaryote and prokaryote species are presented in the mid part of the figure. The pathways already described in the literature are highlighted with bold colors according to the taxonomic groups mentioned, while the putative VC pathways are presented in the gridline sections. The enzymes that intervene in the substrate conversions are presented by numbered circles according to the following scheme: Circles in white represent enzymatic reactions described in the literature, circles in yellow represent hypothetical enzymatic reactions and circles in green indicate novel enzymatic constituents of putative alternative VC synthesis pathways with available functional characterization. The question marks (?) represent putative enzymatic reactions for which no candidate enzyme can be suggested with current data. The tissues correlated with VC synthesis in animal species are represented below the corresponding proposed pathways when available. The dashed box highlights the myo-Inositol VC synthesis pathway that is exclusively used by species that synthesize VC in the kidney. The phylogenetic relationship inferred between the studied AOs is summarized in a cladogram format at the bottom. The enzyme number and substrate acronym lists are presented below the figure to facilitate the interpretation.

**Table 1 genes-13-01917-t001:** Amino acid regions relevant for AO functions in prokaryote and eukaryote species.

Protein Sequences	FAD-Binding Motif (X[GR]X[GS][HLK][SG]) *	HWXK Motif	EXR Pair	Oxygen Reactivity Marker (G)	Conserved Cysteine (C)
*M. tuberculosis H37Rv* bGUDH NP_216287.1	VGS-G**H**S	**HW**G**K**	**E**V**R**	**G**	R
*Pseudomonas aeruginosa* AO NP_254014.1	VGS-G**H**S	**HW**G**K**	**E**Y**R**	A	R
*B. bacterium* AO MBS1617249.1 **	VGT-G**H**S	**HW**G**K**	**E**I**R**	**G**	D
*C. parasitica NIES 267* AO BAY85903.1 **	FGS-G**H**S	**HW**A**K**	**E**V**R**	**G**	**C**
*T. brucei brucei TREU927* GAL XP_844937.1	VGG-GKS	**HW**A**K**	**E**F**R**	P	**C**
*M. musculus* GULO NP_848862.1	VGG-G**H**S	**HW**A**K**	**E**V**R**	**G**	**C**
*A. thaliana* GALDH NP_190376.1	VGS-GLS	**HW**A**K**	**E**Q**R**	A	**C**
*A. thaliana* pGUDH NP_182197.1	VTTRYSH	**HW**G**K**	LM**R**	P	**C**
*Asgard group archaeon* AO MCP8715902.1 **	VGS-G**H**S	**HW**A**K**	**E**V**R**	**G**	**C**
*S. cerevisiae S288C* ALO NP_013624.1	VGS-G**H**S	**HW**A**K**	**E**V**R**	**G**	**C**
*Euglena gracilis* GALDH BAO94255.2	AGA-MLS	**HW**A**K**	**E**Q**R**	S	**C**
*Aspergillus**chevalieri* bGUDH XP_043131130.1	VGN-G**H**G	**HW**N**K**	QF**R**	G	E
*C. lupini* pGUDH XP_049145240.1	SGK-G**H**M	**HW**T**K**	NIK	L	V
*B. mori* DHCR24 XP_004926865.1	CTA-RPT	LYAD	FKI	P	K

* N-terminal FAD-binding motif derived from evidence in [202]. ** Novel prokaryote AOs inferred from the phylogenetic study. Essential amino acids are highlighted in bold, and amino acid regions that do not follow the motifs are underlined.

## Data Availability

Not applicable.

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
