# Peer review of "Advances in Novel Animal Vitamin C Biosynthesis Pathways and the Role of Prokaryote-Based Inferences to Understand Their Origin"

_genes, 2022, doi:10.3390/genes13101917_

Round 1

Reviewer 1 Report

This is a "review" article on an interesting topic, although it isn't purely a review article. I would argue that enough analyses (remote homology detection, phylogenetics) is performed that a methods section is necessary.

I have a few additional comments.

-There are much more sensitive methods to detect remote (distant) homologs than BLASTp.

-Several big picture trees should be shown:

-Section 4.1 presents several alternative hypotheses. These should be depicted with a tree figure.

-A species tree showing what enzymes are present where in the eukaryotic tree of life and where they change on internal branches would be really useful. This should include lineage-specific loss (as in some mammalian lineages).

-Along those lines, a section on variation across mammalian lineages and any associated evolutionary analysis would be interesting to readers.

With several of these points, this will be an interesting and useful contribution to the scientific literature.

Author Response

Reviewer 1

Comment 1. This is a "review" article on an interesting topic, although it isn't purely a review article. I would argue that enough analyses (remote homology detection, phylogenetics) is performed that a methods section is necessary.

We believe that a review article may report the results of simple analyses, based on already publicly available data, if they give insight into the subject being discussed and point to future likely productive avenues of research. A materials and methods section was added after the conclusion section of the manuscript.

Comment 2. There are much more sensitive methods to detect remote (distant) homologs than BLASTp.

Although it is conceivable that the use of much more sensitive methods than BLASTp may provide new insights into the evolution of Vitamin C biosynthesis pathways, we feel that the discussion on the added value of the use of such methods should be provided in a research article, not in a review article, but we now acknowledge this possibility (“…as well as the use of more sensitive methods to detect remote (distant) homologs than BLASTp …”. Being a review, we have used only simple, previously used methods. The BLASTp methodology was already used in Wheeler et al. (2015) to explore the evolution of Vitamin C biosynthesis pathways. This is mentioned in the following sentences: “In [94], the authors state that an extensive homology search led to the inference of only one significant GULO homolog in prokaryote species. The initial search was performed using the M. musculus GULO and A. thaliana GALDH sequences as query in a “blastp” approach [94].”.

Comment 3. Several big picture trees should be shown:

  • Section 4.1 presents several alternative hypotheses. These should be depicted with a tree figure.

The section was rewritten to clarify the hypotheses, and Figures 3 and 4 were added as comprehensive schematic representations.

  • A species tree showing what enzymes are present where in the eukaryotic tree of life and where they change on internal branches would be really useful. This should include lineage-specific loss (as in some mammalian lineages).

A cladogram depicting the presence/absence patterns of GULO/GALDH lineages across eukaryotes, based on the available literature, was added as Figure 3.

  • Along those lines, a section on variation across mammalian lineages and any associated evolutionary analysis would be interesting to readers.

Since the focus of this review is on the novel Vitamin C biosynthesis pathways that are currently being characterized, and none of them was found in mammals, we feel that such discussion is out of the scope of this article. Nevertheless, the information presented in Figure 3 and the corresponding legend provides an accurate summary of the evolutionary history of GULO across mammalian lineages. In the figure legend the reader can find the most relevant references on this subject.

Reviewer 2 Report

The authors summarized current development of animal vitamin C biosynthetic pathways. The authors discussed current VC biosynthetic pathway, and future potential VC biosynthetic using protostomian species or prokaryote species were proposed. The manuscript is interesting and timely. However, the description in the manuscript makes me hard to follow, and several issues need to be addressed.

The title should be revised to “Advance on …“. Many part are unrelated with animal VC biosynthetic pathways. Thus, the authors need to revise the title to fit the manuscript content.

The abstract should be revised to strengthen the study.

The description of line 67 should be revised. Especially, the fungi are the sister group of animal should be removed.

Line 123, in Figure 1 should be move to the last word of the sentence (Figure 1). Many similar expression are oral English, not scientific article. The authors need to revise them.

Line 161-162 “These findings will be discussed in detail in the following sections.” Should be removed.

For the part 4 VC synthesis in bacteria: an unexplored concept. This is different with the title novel animal VC biosynthetic pathways. To my limited knowledge, most VC in the market were produced by fermentation with Ketogulonogenium vulgarum, this unexplored concept should be revised. Moreover, the yield of VC in current industrial process is very high, thereby the authors need to describe details in the review, especially the industrial process for Vc production. Besides, If the authors proposed to improve the industrial production of VC, the application of cutting-edge technologies in VC production, such as synthetic biology and accurate fermentation technology, should be discussed. The authors should give details about how to improve VC production in industry.

For the conclusion part, it should be the summary of the manuscript, however, some parts were not the summarized results. This part should be further improved.

Author Response

Reviewer 2

Comment 1. The authors summarized current development of animal vitamin C biosynthetic pathways. The authors discussed current VC biosynthetic pathway, and future potential VC biosynthetic using protostomian species or prokaryote species were proposed. The manuscript is interesting and timely. However, the description in the manuscript makes me hard to follow, and several issues need to be addressed.

All Reviewer suggestions were taken into account, namely i) several sections were edited according to the suggestions; ii) Figures 3 and 4 were added to clarify the hypotheses presented in section 4.1 and improve readability.

Comment 2. The title should be revised to “Advance on …“. Many part are unrelated with animal VC biosynthetic pathways. Thus, the authors need to revise the title to fit the manuscript content.

The title was changed to “Advances on novel animal Vitamin C biosynthesis pathways and the role of prokaryote based inferences to understand their origin” to more adequately describe the content of the manuscript.

Comment 3. The abstract should be revised to strengthen the study.

The abstract was rewritten to better illustrate the content of the manuscript.

Comment 4. The description of line 67 should be revised. Especially, the fungi are the sister group of animal should be removed.

The sentence was changed to: “Metazoa and Fungi share a close evolutionary relationship and represent the two major groups of the eukaryotic Opisthokonta [58,59]. Nevertheless, in Fungi, knowledge on VC functions is relatively scarce”.

Comment 5. Line 123, in Figure 1 should be move to the last word of the sentence (Figure 1). Many similar expression are oral English, not scientific article. The authors need to revise them.

The sentence was changed according to the suggestion. Similar instances in the remaining manuscript were also changed.

Comment 6. Line 161-162 “These findings will be discussed in detail in the following sections.” Should be removed.

The sentence was removed.

Comment 7. For the part 4 VC synthesis in bacteria: an unexplored concept. This is different with the title novel animal VC biosynthetic pathways. To my limited knowledge, most VC in the market were produced by fermentation with Ketogulonogenium vulgarum, this unexplored concept should be revised. Moreover, the yield of VC in current industrial process is very high, thereby the authors need to describe details in the review, especially the industrial process for Vc production. Besides, If the authors proposed to improve the industrial production of VC, the application of cutting-edge technologies in VC production, such as synthetic biology and accurate fermentation technology, should be discussed. The authors should give details about how to improve VC production in industry.

The title of the section was changed to “Prokaryote AOs: an unexplored concept” to better depict the content of the following subsections. Due to the exploratory nature of the results, no concrete industrial application can be discussed in the context of the findings that are presented. However, a brief discussion on the current status of industrial VC synthesis and possible implications of further studies of prokaryote AOs to the optimization of the process was added to section 4.2.

Comment 8. For the conclusion part, it should be the summary of the manuscript, however, some parts were not the summarized results. This part should be further improved.

The conclusion was rewritten to accurately summarize the content of the manuscript.

Round 2

Reviewer 1 Report

My one remaining comment is that the events that are depicted at nodes in the tree in Figure 3 really happened on the branch leading to that node (nodes are points in time of zero length). If the figure is not to be modified, the legend should be modified to indicate this.

Author Response

Reviewer 1

Comment 1. My one remaining comment is that the events that are depicted at nodes in the tree in Figure 3 really happened on the branch leading to that node (nodes are points in time of zero length). If the figure is not to be modified, the legend should be modified to indicate this.

The legend of Figure 3 was modified according to the suggestion.

Reviewer 2 Report

The authors still need further revision. 

Synthetic biology used to synthesize VC should be discussed.

Moreover, part 6. Materials and Methods should be part of the figure legend or move to the supplementary file 

Author Response

Reviewer 2

Comment 1. Synthetic biology used to synthesize VC should be discussed.

A more extensive discussion regarding synthetic biology methodologies applied to the development of one-step fermentation alternatives for industrial VC synthesis was included in section 4.2. The potential implications of phylogenetic inferences regarding prokaryotes to the development of industrial VC synthesis solutions are also discussed, focusing primarily on an example regarding Pseudomonas aeruginosa.

Comment 2. Moreover, part 6. Materials and Methods should be part of the figure legend or move to the supplementary file.

The Materials and Methods section was edited and added to the legend of Figure 5 according to the suggestion.